# REINFORCEMENT LEARNING FOR CLINICAL REASONING: ALIGNING LLMS WITH ACR IMAGING APPROPRIATENESS CRITERIA

## ABSTRACT

Medical imaging has revolutionized diagnosis, yet unnecessary procedures are rising, exposing patients to radiation and stress, limiting equitable access, and straining healthcare systems. The American College of Radiology Appropriateness Criteria®, developed through extensive multidisciplinary review, provide evidence-based guidance but remain underutilized. Leveraging advances in LLM reasoning, we introduce a Reasoning Agent trained with Reinforcement Learning (RL), specifically Group Relative Policy Optimization (GRPO), to replicate expert clinical reasoning from the ACR Criteria. We present a novel RL approach for structured medical reasoning, systematically comparing reasoning-focused reward functions and evidence integration strategies. Our lightweight 8B model, *MedReason-Embed*, improves macro F1 by 18% over baseline, shows stronger reasoning alignment, and outperforms both larger and alternatively trained models, showing that reasoning-based supervision enables efficient, trustworthy clinical AI. Building on this, we design a modular end-to-end agentic architecture that automates imaging referrals: mapping diagnoses to ICD codes, retrieving PubMed evidence, and recommending optimal procedures. Crucially, the ability to generalize beyond static ACR guidelines not only enables clinicians to handle out-of-distribution cases, but also supports scaling the guideline development process itself, potentially reducing the significant effort required to create and update them. This work shows the potential of reasoning-focused RL within agentic architectures to deliver transparent, scalable, and reliable clinical decision support. Our code is available at: `https://anonymous.4open.science/r/agentic-imaging-recommender-iclr-877D`

## 1 INTRODUCTION

Low-value medical imaging, defined as procedures whose risks outweigh their benefits, has risen sharply in recent years. Unnecessary CT and MRI use is particularly concerning, with 20–50% of CT scans in the US deemed unnecessary U.S. Food and Drug Administration (2022), and other studies reporting 35–80% of imaging outside established clinical standards Alahmad et al. (2024); Deshommes et al. (2024); Lavery et al. (2024); Marin et al. (2024). Such procedures expose patients to harmful radiation linked to increased cancer risk, contribute to overdiagnosis and anxiety, delay critical diagnoses, and waste healthcare resources Miglioretti et al. (2013), with reductions estimated to save approximately $12 billion annually in the US Radiology Business (2025).

To reduce inappropriate imaging, the American College of Radiology (ACR) developed the Appropriateness Criteria® (ACR-AC), evidence-based guidelines for imaging selection of Radiology (2025). For nearly 30 years, expert panels systematically review literature for clinical scenarios, apply the GRADE scale to rate evidence quality Schünemann et al. (2008), and synthesize this evidence into graded recommendations balancing diagnostic value and patient risk Kurth et al. (2021). As of September 2025, they cover 257 imaging topics, yet adoption is very low, with fewer than 1% of clinicians using them as a primary reference Bautista et al. (2009).

AI systems, particularly Large Language Models (LLMs), hold promise for guideline-based decision support but face key barriers, including hallucinations, shortcut learning, and limited explainability

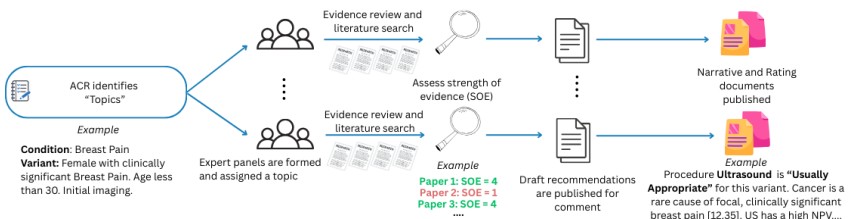

Figure 1: ACR-AC Guideline Development Process.

that challenge clinical validity and regulatory compliance Pan et al. (2025). Although recent work shifts LLMs toward stepwise "System 2" reasoning Li et al. (2025), medical LLMs remain largely trained with Supervised Fine-Tuning (SFT), that often leads to weak reasoning Chu et al. (2025). Reinforcement learning (RL) offers a stronger alternative, using reward signals to improve reasoning, scalability, and generalization, with Group Relative Policy Optimization (GRPO) Guo et al. (2025) providing scalable rule-based reward comparisons and early success across domains including medicine Jaech et al. (2024). While RL effectiveness depends heavily on reward design, most medical RL models use rewards for format compliance and final correctness often optimizes only format or correctness, with some studies show these fail to capture the complexity of clinical reasoning DeYoung et al. (2019).Incorporating reasoning steps yields stronger rewards than answer-only signals Cobbe et al. (2021), yet remains underexplored in medicine. We address this by leveraging ACR-AC expert knowledge to design rewards that align intermediate reasoning with clinical standards in imaging referrals.

Moreover, grounding in high-quality literature improves accuracy and trust Wang et al. (2024), and, importantly, enables generalization beyond static, human-dependent guidelines. Building on this, we designed a deployable architecture that mirrors the the ACR-AC development process; but replaces time-limited human experts with LLM agents. It automatically retrieves and filters studies from medical databases, ensuring recommendations are based on strong, up-to-date evidence.

Starting from a clinical note, the system standardizes it and maps it to the International Classification of Diseases, Ninth Revision coding system (ICD-9-CM) U.S. Department of Health and Human Services (2011), ensuring interoperability with hospital workflows. It then retrieves and filters medical literature, synthesizing the evidence into a final imaging recommendation using a GRPO-trained Reasoning Agent. This agent outputs both an appropriateness recommendation and a concise, evidence-grounded justification. We align intermediate reasoning with expert ACR-AC traces through custom rewards, going beyond answer-level optimization. To our knowledge, this is the first medical study to apply RL with reasoning supervision and compare reasoning-focused rewards, advancing decision support toward greater transparency and trust. By closely mirroring the ACR workflow, the system notably also supports a capability unmatched by other systems: providing justified recommendations even for conditions not covered by the ACR-AC, that though comprehensive, are not exhaustive Chan et al. (2019).

Our main contributions are: (a) we design an end-to-end agentic architecture for clinical imaging referrals that integrates ICD coding, evidence retrieval, and GRPO-trained reasoning, closely replicating the ACR decision process while enabling generalization beyond fixed guidelines; (b) we introduce and evaluate GRPO-adapted reasoning models, showing that RL lightweight models can rival larger ones by providing accurate, transparent imaging recommendations; (c)we contribute reasoning-focused rewards aligned with ACR traces and evidence-integration strategies, improving performance, reasoning alignment, and clinical reliability.

## 2 BACKGROUND AND RELATED WORK

### 2.1 ICD CODING

To align with clinical practices, we use the International Classification of Diseases (ICD) for Health Statistics (US) the WHO-maintained global standard that hierarchically organizes conditions with standardized codes and descriptions (e.g., ICD-9-CM code 611.71 "Mastodynia" under 611 "Breast

Disorders").Our system maps notes to ICD codes as an *intermediate layer* before linking to ACR guidelines, projecting both notes and ACR conditions onto a shared validated vocabulary to reduce errors and ensure interoperability.

ICD coding, the assignment of standardized codes to clinical documentation, is an important task, as evidenced by commercial adoption in tools like Ambience Ambience Healthcare (2025), but remains challenging due to severe label imbalance and variation in note style, and language Zhou et al. (2021). Manual coding is slow and error-prone, with widely variable accuracy ranging from 50% to 98% in the UK Burns et al. (2012), motivating automation. Methods have evolved from rule-based approaches Farkas & Szarvas (2008), to embeddings Gomes et al. (2024); Michalopoulos et al. (2022) and neural networks Azam et al. (2019), often using code hierarchies to improve accuracy Chen & Ren (2019). LLMs show promise for ICD coding but often hallucinate and underperform on rare cases, reaching only ∼46% top-1 accuracy Mustafa et al. (2025); Soroush et al. (2023). Retrieval-Augmented Generation and reranking approaches show improved results, with MedCoder achieving 0.60 F1 score on synthetic data Baksi et al. (2024). Still, performance remains modest and variable; perhaps highlighting the need for tailored models, validation, and human oversight Abdelgadir et al. (2024).

## 2.2 CLINICAL REASONING MODELS

LLMs are shifting from "System 1" thinking to analytical "System 2" reasoning Li et al. (2025), as seen in models like OpenAI's o1 and DeepSeek's R1 Jaech et al. (2024). This is especially important in medicine, where safe deployment requires transparent reasoning. Supervised Fine-Tuning (SFT) has been the dominant post-training method but studies have shown that it often leads to shortcut learning and poor generalization since it only encourages reasoning implicitly Chu et al. (2025). Reinforcement Learning (RL), by contrast, optimizes models with task-specific rewards and better supports complex objectives like multi-step reasoning. Recent methods such as GRPO and its variants like Dr. GRPO Liu et al. (2025), and Group Sequence Policy Optimization (GSPO)Zheng et al. (2025), enabled material improvements in efficiency and reasoning.

RL methods improve reasoning and show clear benefits in medical tasks, even with limited data. MedVLM-R1, a 2B-parameter model trained on only 600 samples, gained 20% accuracy and showed strong generalization on medical visual question answering (VQA) benchmarks Pan et al. (2025), while Med-R1 achieved nearly 30% gains across eight modalities, even outperforming a much larger counterpart Lai et al. (2025). Across studies, RL-adapted models consistently outperform SFT, especially in generalization Lai et al. (2025), with a study summarizing this as "SFT memorizes, RL generalizes" Chu et al. (2025). Currently, most RL medical models rely on dual rewards for answer correctness and formatting, aiming to produce accurate and structured outputs. Studies show that generic rewards often fail to capture true clinical reasoning Chen et al. (2025), and Pan et al. (2025) further note that correct answers can sometimes mask flawed reasoning.

Reward functions are central to RL. While many argue that clinical reasoning should be the primary reward Brodeur et al. (2024), most works still optimize answer accuracy Chen et al. (2025).A key design choice is the supervision strategy of the reward model; Outcome-supervised Reward Models (ORMs) reward the final answer, while Process-supervised Reward Models (PRMs) evaluate intermediate steps Lightman et al. (2023). PRMs may reward identifying clinical risks before recommending imaging, helping the model learn the reasoning process rather than just the outcome, like an ORM would. PRMs encourage interpretable, aligned reasoning, reduce errors and reward hacking, and have been shown to outperform ORMs in multi-step reasoning Amodei et al. (2016).

Despite these benefits, PRMs remain underused in medicine mainly due to the lack of a clear framework for evaluating reasoning. Surface metrics based on n-gram overlap such as BLEU miss semantics Schluter (2017), while semantic approaches like verifier training requires costly annotations Li et al. (2022); Lightman et al. (2023). A growing alternative is LLMs-as-critics, which assess reasoning for faithfulness and coverage Gu et al. (2024). While models using this approach, such as LlamaV-o1 Thawakar et al. (2025) and PathVLM-R1 Wu et al. (2025) improve on reasoning, these evaluators come at a high computational cost.

In summary, a pressing need exists in the biomedical space to distill clinical reasoning for agents handling medical referrals. Building on the advantages of PRMs over ORMs Lu et al. (2024); Zhu et al. (2025) and the clinician-trust reasoning fosters Brodeur et al. (2024), we propose and evaluate

multiple reward functions that compare generated reasoning with gold-standard reasoning from the ACR-AC to encourage models to learn expert-like thinking.

## 2.3 MEDICAL RETRIEVAL

Clinical decision support requires expert-aligned reasoning grounded in solid evidence Wang et al. (2024). Our approach combines reasoning rewards with literature retrieval to reduce hallucinations even without in-domain training Tan et al. (2025). RetCare Wang et al. (2024) showed that grounding outputs in PubMed's 38 million citations pub enhances accuracy and clinician trust, though it is generally accepted that its keyword-based retrieval could be further improved. Effective evidence retrieval is difficult given the rapid growth of medical literature Lu (2011). Beyond query expansion and filters Lu (2011), DeepRetrieval offers a recent innovative solution Jiang et al. (2025). By training an LLM with RL to rewrite queries into Boolean logic, it boosts recall to 65% versus 25% with LEADS Wang et al. (2025). For example, the query "best imaging for breast pain" becomes "((Breast Pain OR Mastalgia) AND (Imaging OR Mammography OR MRI OR CT))", expanding with synonyms to improve retrieval.

However, an effective evidence-search strategy alone is not enough; filtering high-quality results remains expert-dependent and time-consuming Abdelkader et al. (2021). Machine learning (ML) approaches have been proposed to predict evidence quality using features like study design, sample size, and other metadata, reaching 60–70% accuracy Abdelkader et al. (2021), though comparisons are difficult due to differing methods. Still, ML automation remains a promising option for strengthening evidence assessment.

Our system uses DeepRetrieval Jiang et al. (2025) to query PubMed for candidate papers, which are then filtered for quality. We rely on this off-the-shelf retriever to adequately handle large-scale biomedical search, enabling us to focus on the medical reasoning.

## 3 METHODOLOGY

### 3.1 SYSTEM ARCHITECTURE

Our system employs a modular, agent-based architecture, where outputs are passed forward in a pseudo-sequential flow to mirror clinical reasoning, orchestrated via LangGraph.

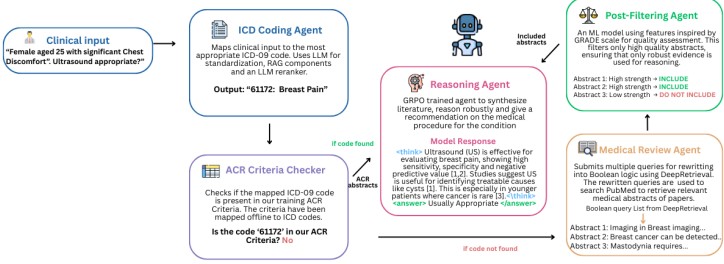

Figure 2: Overview of the multi-agent system architecture.

Our cognitive architecture is designed to address the question: 'For a patient with condition Y, is procedure X appropriate?'. Firstly, the ICD Coding Agent maps clinical diagnostic notes to ICD-9-CM codes, which the ACR Criteria Checker uses to match against the ACR-AC list. If the mapped ICD code matches the ACR list, the relevant guideline evidence is passed to the Reasoning Agent; otherwise, the Medical Review Agent retrieves relevant studies literature interrogating PubMed using DeepRetrieval-mediated queries, and the Post-Filtering Agent selects high-quality evidence using GRADE principles. Finally, the Reasoning Agent pieces together the evidence into a recommendation for or against a particular medical imaging procedure. Implementation details for each agent are provided in the following sections.

## 3.2 ICD CODING AGENT

We developed our ICD coding Agent on a synthetic dataset of ~6,500 records, each containing an Italian natural language diagnostic note and its associated ICD-9-CM code, covering diverse codes, scenarios, and notation styles. Quality was improved using embedding-based similarity to detect and correct mislabeled or ambiguous entries, yielding a reliable dataset. Our dataset contains short, less ambiguous notes with one code per record, highlighting the data-specific nature of ICD coding. We then applied this approach to both clinical notes and ACR-AC conditions, creating an *ICD intermediate space* for easier alignment and deployment.

To improve ICD coding, clinical texts were first standardized into English medical language using LLaMA-3.1-8B Grattafiori et al. (2024), effectively reducing semantic mismatch, improving alignment with ICD terminology and retrieval. Standardized texts were then matched to ICD code descriptions found in our dataset with FAISS-based similarity search, while the same LLM reranked results whenever retrieval showed ambiguity, including low similarity or high variance among top candidates. Performance was evaluated with top-1/top-5 accuracy, hierarchical accuracy defined as a 3-digit match indicating if the key condition is captured, and Mean Reciprocal Rank (MRR). This novel hybrid pipeline combines LLM standardization, dense retrieval, and reranking; facilitating deployment and providing a bridge between clinical notes and the guideline criteria.

## 3.3 REASONING AGENT

**Preprocessing the ACR Criteria**
The *Reasoning Agent* was developed from a subset of 30 different ACR-AC conditions, covering varied categories, body regions, and symptoms. From each, we parsed their "Narrative and Rating" documents to obtain condition–variant–procedure triplets with their corresponding appropriateness ratings ("Usually Appropriate", "May Be Appropriate", or "Usually Not Appropriate") and expert-authored justification texts, yielding ~1,800 entries after excluding non-consensus cases.

**Extracting reasoning traces**
ACR justifications are often lengthy and complex, so we extracted concise "reasoning traces": atomic, verifiable claims capturing the core rationale, following Huang et al. (2025). We used LLaMA-4-Scout-17B-16E-Instruct for its strong summarization and reasoning skills AI (2025a) at the time of writing, with human oversight to prevent hallucinations or omissions. This standardized format improves interpretability, supports downstream tasks, and eases expert review, as ACR-AC reasoning is usually unordered factual points rather than complex logical chains.

**Group Relative Policy Optimization (GRPO)**
This agent is trained with GRPO and LoRA-based fine-tuning Hu et al. (2022) using the Unsloth framework Daniel Han & team (2023). GRPO, introduced in DeepSeekMath Shao et al. (2024), extends Proximal Policy Optimization (PPO) Schulman et al. (2017) by removing the reward value network. For each prompt $q$, the model generates $G$ outputs $\{o_i\}_{i=1}^{G}$, each assigned a reward $r_i$, and advantages are computed relative to the group mean, simplifying training and lowering cost. The clipped PPO-style objective with importance ratio $\rho_i$ ensures stability, while a KL penalty keeps the new policy $\pi_{\theta_{\text{new}}}$ close to a frozen reference $\pi_{\text{ref}}$. The objective maximizes expected clipped advantage across outputs while minimizing KL divergence.

$$\mathcal{J}_{\text{GRPO}}(\theta) = \mathbb{E}_{q,\{o_i\}} \left[ \frac{1}{G} \sum_{i=1}^{G} \min\big(r_i\rho_i,\, \text{clip}(\rho_i, 1-\epsilon, 1+\epsilon)r_i\big) - \beta\, \mathbb{D}_{\text{KL}}(\pi_{\theta_{\text{new}}} \| \pi_{\text{ref}}) \right] \quad (1)$$

**Model Variants and Reward Designs**
We evaluate multiple aspects of the *Reasoning Agent*, which is built on the LLaMA-3.1-8B backbone, chosen for its open-source availability at the time this study begun, size, and compatibility with RL training frameworks like Unsloth. Model variants and rewards are described below and implementation details and a training example can be found in Appendix A.

**Baseline model**: Our first model was trained using standard rewards common in prior work: one binary "answer" reward for the correct appropriateness label and another binary "format" reward for properly enclosing reasoning within `<think></think>` followed by the answer in `<answer></answer>` tags.

**Citations model**: Prior RL-based medical models often over-rely on on the backbone's pre-trained knowledge, risking outdated or inaccurate outputs. The *Citations* model tests whether grounding in ACR-AC cited evidence improves reliability. References were retrieved via PubMed IDs, with abstracts condensed into results/conclusions subsections and added as context. The model uses the same rewards as the *Baseline*, but with condensed abstracts as additional input.

**LLM Eval model**: Building on *Citations*, this model adds an LLM-based reward inspired by Thawakar et al. (2025); Wu et al. (2025), that scores generated reasoning against ACR-AC gold, explicitly teaching what constitutes "good" reasoning, encouraging closer alignment. A smaller Qwen1.5-1.8B evaluator Qwen Team (2024) judges concept overlap, logical consistency, and evidence use, yielding a continuous score between 0 and 1.

**MedReason-Embed model**: This model introduces a novel *joint reward*, motivated by findings that models may give correct answers with flawed reasoning Chen et al. (2025). For each gold reasoning sentence, we take its maximum cosine similarity with generated sentences, average these scores, and multiply by a binary answer reward; rewarding the model when its answer is correct. The design links reasoning quality to outcome correctness, enforcing that *valid reasoning must lead to the correct decision*. While it ignores good reasoning when paired with wrong answers, it helps with risk–benefit assessment, improving both reasoning and accuracy. The *MedReason-Embed* therefore uses two rewards: the binary format reward and this joint reasoning reward, with all medical evidence included. Mathematically, it is defined as: $R_{\text{joint}} = \mathbb{I}_{\text{gen=gold}} \cdot \frac{1}{N} \sum_{i=1}^{N} \max_j \cos(\mathbf{e}_i^{\text{gold}}, \mathbf{e}_j^{\text{gen}})$.

Table 1: Comparison of Reasoning Agent Variants

| Model | Evidence | $r_{answer}$ | $r_{format}$ | $r_{reasoning}$ |
|---|---|---|---|---|
| *Baseline* | None | ✓ | ✓ | None |
| *Citations* | ✓ | ✓ | ✓ | None |
| *LLM Eval* | ✓ | ✓ | ✓ | LLM-based |
| *MedReason-Embed* | ✓ | ✓ (via reasoning) | ✓ | Joint reward |

Finally, in line with prior work Lai et al. (2025), we compared our RL-based models to standard approaches. We trained an SFT model on the same dataset with the LLaMA-3.1-8B backbone, cross-entropy loss, and matched parameters for fairness. We also evaluated the larger LLaMA-3.1-405B AI (2025b), used in raw form without task-specific fine-tuning. Both baselines predict only appropriateness labels, without reasoning, as in prior work Pan et al. (2025).

**Evaluation of Reasoning Agent**
We evaluate models on three aspects: predictive performance, reasoning alignment, and training efficiency. For prediction, due to dataset imbalance (∼64% "Usually Not Appropriate"), we report Macro F1 to weight classes equally and highlight minority performance, and Weighted F1 to reflect prevalence. Reasoning alignment is assessed by two metrics: (i) an LLM-based score (*LLM-align-score*) rating *clinical relevance* and *medical knowledge* on a 0–10 scale following Zhu et al. (2025), and (ii) a NER-based F1 score using OpenMed's pipeline OpenMed (2024), comparing embedded entities and rule-based phrases between expert and generated reasoning to capture semantic overlap of clinically meaningful concepts while avoiding hallucinations. Finally, training efficiency is reported relative to the baseline, since inference time is similar across models.

### 3.4 MEDICAL REVIEW AND POST-FILTERING FOR GENERALIZATION

For the *Medical Review Agent*, we use the Deep-Retrieval-PubMed-3B model DeepRetrieval Team (2024). Because a single query rarely captures all relevant aspects of the ACR-AC strategy (e.g., synonyms, related conditions, procedure terms), we submit multiple rewritten queries per condition (see Appendix B). We focus on retrieving evidence with similar *clinical concepts* to ACR-AC references to approximate their reasoning, as replicating their extensive manual review is infeasible. Coverage is evaluated by comparing retrieved evidence against ACR-AC references using clustered embeddings and topic modeling (LDA) Blei et al. (2003). Retrieved evidence is passed to the *Post-Filtering Agent*, which models the GRADE scale Schünemann et al. (2008) to score strength of evidence (SOE) and prioritize quality studies. ML models trained on ACR-AC SOE labels using PubMed metadata (e.g., publication type) and abstract features (e.g., cohort size) are evaluated mainly on recall of "high" SOE papers to ensure inclusion of top-tier studies.

Our final goal is to test generalization. We created a set of four unseen conditions (Appendix 8), chosen for clinical relevance and low similarity to the training set, spanning diverse regions, domains, and diagnostic purposes. For these cases, the full pipeline of evidence retrieval was run to simulate performance outside the ACR-AC. We need to evaluate how well the model handles both unfamiliar scenarios and alternative evidence sources. Specifically, (a) we test the models on new conditions, using ACR-AC citations as reference to isolate condition novelty, and (b) we vary the evidence source, comparing ACR-AC references with our retrieved literature on the same conditions.

## 4 RESULTS

### 4.1 ICD CODING

Table 2 shows the performance of the *ICD Coding Agent*. The LLM standardization step proved effective, mapping the original clinical notes to ICD descriptions with near-exact wording and handling language variability without heavy hallucinations.

Table 2: Evaluation Results ICD Agent

| Metric | Value |
|--------|-------|
| Top-1 Accuracy | 80.45% |
| Top-5 Accuracy | 91.97% |
| Mean Reciprocal Rank (MRR) | 85.46% |
| Top-1 Hierarchical Accuracy | 91.47% |

Results show reliable performance: top-1 accuracy of 80.45%, hierarchical accuracy of 91.47% showing that the key condition is captured in the vast majority of cases, and MRR of 85.46%, with the correct code in the top-5 nearly 92% of the time. The LLM reranker added a modest 0.5% boost. While promising, results may reflect the dataset's relatively short and unambiguous notes.

### 4.2 MODEL PERFORMANCE AND REASONING QUALITY

The Reasoning Agent is trained on 1,800 condition-variant-procedure triplets from 30 conditions, with stratified 70/30 train-test split. Table 3 compares the four RL models, the SFT, and the larger LLaMA-3.1-405B, while Table 4 reports reasoning quality and training metrics for the RL models.

Table 3: Model Predictive Performance Results

| Model / Config | Macro Avg F1 | Weighted F1 |
|----------------|--------------|-------------|
| Baseline | 33.5% | 37.1% |
| Citations | 45.6% | 56.6% |
| LLM Eval | 52.7% | 65.6% |
| MedReason-Embed | 51.6% | 65.6% |
| SFT model | 36.7% | 56.2% |
| LLaMA 405B | 47.0% | 53.7% |

Table 4: Model Reasoning Alignment and Training efficiency

| Model/Config | LLM-align-score (/10) | NER Embedding F1 | Relative Training Time |
|--------------|----------------------|------------------|------------------------|
| Baseline | 5.64 | 37.9% | 1.0 |
| Citations | 7.28 | 61.2% | 1.2 |
| LLM Eval | 7.57 | 65.4% | 1.8 |
| MedReason-Embed | 7.67 | 65.5% | 1.3 |

The *Baseline*, trained only with format and answer rewards as in most literature, achieves macro and weighted F1 scores of 33.5% and 37.1%, scores not far off from a majority-class classifier, showing that while the *Baseline* captures class diversity, it remains weak overall. Reasoning alignment is also low (5.64/10; 37.9%), making it unsuitable for deployment. Adding medical evidence in the *Citations* model substantially improves performance (45.6% macro F1, 56.6% weighted F1), a gain

of +12% and +20% over the *Baseline*, alongside higher reasoning alignment (7.28/10; 61.2%). This highlights that high-quality context enables foundation models to adapt effectively to domain tasks.

Using reasoning rewards further boosts performance. The *LLM Eval* model achieves 52.7% macro and 65.6% weighted F1, with stronger alignment (7.57/10; 65.4%), but at high cost since each generated answer requires an extra LLM call. Our *MedReason-Embed* model reaches similar scores (51.6%, 65.6%) at much lower cost, showing that task-specific rewards can match resource-intensive strategies. Overall, rewarding reasoning quality improves both alignment and answer accuracy. McNemar's test McNemar (1947) confirmed significant differences between all models except *LLM Eval* and *MedReason-Embed*, which perform comparably at very different computational costs.

Lastly, the *SFT* model (36.7% macro; 56.2% weighted F1) predicts mostly "Usually Not Appropriate', overfitting to the majority class, while the much larger *LLaMA-3.1-405B* (47.0%; 53.7%) only matches the *Citations* model, lags behind our reasoning-based models, and demands far greater resources. Overall, our RL models with evidence and reasoning rewards outperform both SFT and larger models, showing that domain-specific context and reward design, rather than scale alone, drive robust and efficient reasoning.

### 4.3 EVIDENCE RETRIEVAL, POST-FILTERING AND GENERALIZATION

For the *Medical Review* Agent, we used Deep-Retrieval-PubMed-3B, with the best strategy retrieving 25 papers per condition via structured Boolean queries (Appendix B). For the *Post-Filtering* Agent, a Random Forest Biau & Scornet (2016) leveraging study design, SJR score, and sample size achieved 0.74 recall for high-strength studies, generalized well, with study design and journal quality as top features. The full feature list and task definition are provided in Appendix C.

To directly evaluate our evidence gathering pipeline, we evaluate generalization on four unseen conditions (roughly half the test set), comparing performance with (a) gold ACR-AC citations (Table 5) and (b) our pipeline's citations (Table 6). Results for all RL models, SFT, and LLaMA-3.1-405B are reported against each other and their original test set scores (Table 3).

Table 5: Generalization dataset performance with **ACR citations**

| Model / Config | Macro Avg F1 | Weighted F1 |
| --- | --- | --- |
| Baseline | 31.0% | 34.5% |
| Citations | 40.5% | 53.0% |
| LLM Eval | 46.6% | 63.6% |
| MedReason-Embed | 44.5% | 63.3% |
| SFT | 43.3% | 65.1% |
| LLaMA 405B | 51.7% | 60.0% |

Table 6: Generalization dataset performance with **our own citations**

| Model / Config | Macro Avg F1 | Weighted F1 |
| --- | --- | --- |
| Baseline | 31.0% | 34.5% |
| Citations | 40.4% | 46.6% |
| LLM Eval | 43.8% | 54.9% |
| MedReason-Embed | 45.9% | 55.0% |

Firstly, we compare test and generalization performance using ACR-AC citations, to capture the effect of condition novelty (Tables 3, 5). The *Baseline* dropped only 2.5% in F1, while *Citations*, *LLM Eval*, and *Custom Embedding* fell 5–6% macro and 2–3% weighted F1, showing modest sensitivity to distribution shifts. This likely reflects heterogeneity in the generalization set, but rankings remained stable with no signs of overfitting; such small gaps are expected and suggest potential generalization, consistent with other studies Pan et al. (2025). The SFT model still overpredicted "Usually Not Appropriate", confirming poor generalization, while LLaMA-3.1-405B slightly exceeded its test score but remains impractical due to size.

We next test on the generalization set using our retrieved citations (Table 6) versus ACR-AC citations (Table 5), to assess evidence source effects. This setup tests full autonomy; retrieving, filtering,

and reasoning without human curation. The *Baseline* was unchanged as it uses no citations, while *Citations* held macro F1 ($\sim$40%) but dropped 6% weighted F1, still beating *Baseline* by 9–12% with our citations. *LLM Eval* fell slightly (–3% macro, –8% weighted) yet remained stronger than *Citations*, and *MedReason-Embed* gained 1% macro but lost 8% weighted F1, showing robustness to diverse evidence sources, though further analysis is needed to understand these shifts. Overall, results were comparable across different evidence sets, suggesting autonomy without human-curated citations is feasible, though broader tests are needed.

## 5 Conclusions and Discussion

We introduced a cognitive architecture reproducing the full imaging referral pipeline; from patient condition to ICD coding, evidence retrieval, filtering, and structured reasoning—built for efficiency, scalability, and integration with clinical workflows. Trained on only 30 conditions versus 257 in ACR-AC, it is designed to generalize beyond covered cases, complementing guidelines where none exist, especially since the ACR cannot feasibly cover every scenario. While not a substitute for expert consensus, the system offers a low-cost complement to guideline development, timely amid debates on AI's role in radiology The New York Times. The main conclusions are presented below.

**Strong Performance of Our ICD Coding Agent:** Our Agent, combining LLM-based standardization, RAG, and LLM reranking, achieved a top-1 accuracy of over 80% on real-world data, competitive with recent research systems and in line with average human accuracy in the UK (83%) Burns et al. (2012). These results reinforce the emerging role of *LLMs as standardizers* in clinical NLP Agrawal et al. (2023); Yao et al. (2024) and encourage broader adoption of this pipeline, though further research is needed to assess performance in more complex and ambiguous cases.

**RL for Efficient Clinical Reasoning:** RL effectively adapts general-purpose LLMs for clinical reasoning, outperforming SFT and even larger models. Contextualizing models further boosted performance and alignment, offering a simple strategy to adapt models without fine-tuning. Crucially, reasoning-specific rewards consistently surpassed evidence-only models, improving both performance and reasoning, reinforcing that process-based supervision is more effective than answer-only rewards. *MedReason-Embed* offered the best balance of performance, alignment, and efficiency, showing stable reasoning and signs of self-correction. These results highlight reasoning-aligned RL as a scalable path to trustworthy, lightweight clinical AI.

**Robust Evidence Gathering and Generalization:** Our retrieval and filtering pipeline effectively handled cases without guidelines. DeepRetrieval queries and the *Post-Filtering Agent* reliably surfaced high-quality studies (recall 0.74), offering a scalable alternative to manual review. Performance remained stable across different evidence sources, suggesting robustness. While performance declined slightly on our generalization set of unseen conditions, our two RL reasoning models (*LLM Eval*, *MedReason-Embed*) outperformed SFT under distribution shift, consistent with Pan et al. (2025). While the generalization set was limited, these results suggest our system can extend ACR-AC reasoning with robustness and clinical reliability.

**Our Architecture Enables Scalable Deployment:** The system is built for efficiency and adoption, with ICD coding for hospital workflow alignment, a fast evidence pipeline supporting continuous guideline refinement as new research emerges, and a modular graph design enabling extensions like multi-agent reasoning (e.g. virtual panels), ensuring scalability and flexibility in clinical use.

**Limitations and Future Work**
While reasoning-based rewards improved performance and alignment, embedding- and LLM-based methods may overlook logical errors and clinical nuances. Additional limitations include reliance on abstracts rather than full texts, narrow retrieval from a small number of PubMed papers, and a limited generalization set, which should be expanded for more reliable conclusions about robustness. Future work should broaden evaluations, integrate knowledge graphs (e.g., UMLS Bodenreider (2004), MedGraphRAG Wu et al. (2024)), and explore stronger medical backbones such as MedGEMMA Sellergren et al. (2025), ideally in collaboration with clinical experts. Beyond ACR, the framework could also extend to other guidelines such as NICE National Institute for Health and Care Excellence and ESR European Society of Radiology.

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

# A  REASONING AGENT

## A.1  REASONING AGENT IMPLEMENTATION

All experiments were run on an NVIDIA A100 80GB GPU. We used GRPO with Unsloth Daniel Han & team (2023), which leverages vLLM Kwon et al. (2023) and LoRA Hu et al. (2022) for efficient training Unsloth (2024). Models were trained for 2 epochs, with the Baseline model completing in approximately 2 hours.

Table 7: Reasoning Agent Training Configuration

| | |
|---|---|
| **LoRA Rank** | 32 |
| **Weight Decay** | 0.1 |
| **Warmup Ratio** | 0.1 |
| **LR Scheduler** | Cosine |
| **Optimizer** | `paged_adamw_8bit` |
| **Learning Rate** | $5 \times 10^{-6}$ (AdamW, $\beta_1 = 0.9, \beta_2 = 0.99$) |
| **Batch Size** | 6 |
| **Gradient Accumulation** | 2 steps |
| **Num Generations** | 6 |
| **Max Prompt Length** | 4500 tokens |
| **Max Steps** | 200 (approx. 2 epochs on dataset) |
| **Save Steps** | 100 |
| **Max Grad Norm** | 0.1 |

## A.2  ACR PROCESSING

Table 8: ACR Conditions

| **Train/Test set** | |
|---|---|
| Abnormal Liver Function Tests | Crohn's Disease |
| Abnormal Uterine Bleeding | Dementia |
| Acute Elbow and Forearm Pain | Endometriosis |
| Acute Hip Pain | Female Breast Cancer Screening |
| Acute Nonlocalized Abdominal Pain | Female Infertility |
| Acute Pancreatitis | Head Trauma in Children |
| Acute Shoulder Pain | Headache |
| Acute Spinal Trauma | Hernia |
| Acute Trauma to the Knee | Low Back Pain |
| Anorectal Disease | Male Breast Cancer Screening |
| Back Pain - Child | Osteonecrosis |
| Brain Tumors | Osteoporosis and Bone Mineral Density |
| Breast Pain | Renal Failure |
| Chronic Foot Pain | Scoliosis - Child |
| Congenital or Acquired Heart Disease | Suspected and Known Heart Failure |
| **Generalization test** | |
| Ovarian Cancer Screening | Seizures and Epilepsy |
| Chronic Elbow Pain | Thoracic Back Pain |

## A.3 Reward functions and examples

- **Answer Reward ($r_{\textbf{ans}}$)**: Binary reward for correctly predicting the appropriateness label. *Purpose:* Ensures correct clinical recommendations.

- **Format Reward ($r_{\textbf{fmt}}$)**: Binary reward for using proper `<think>` and `<answer>` tags. *Purpose:* Enforces consistent output formatting to support structured thinking and improve performance.

- **LLM Evaluator Reward ($r_{\textbf{LLM}}$)**: LLM scores reasoning alignment with gold examples (scaled to 0–1 scale), rewarding medically relevant, expert-like reasoning (*LLM-Eval* model).

- **MedReason-Embed Reward ($r_{\textbf{joint}}$)**: Combines answer correctness with reasoning trace alignment (avg. max cosine similarity × binary correctness), promoting both correct answers and well-aligned reasoning (*MedReason-Embed* model).

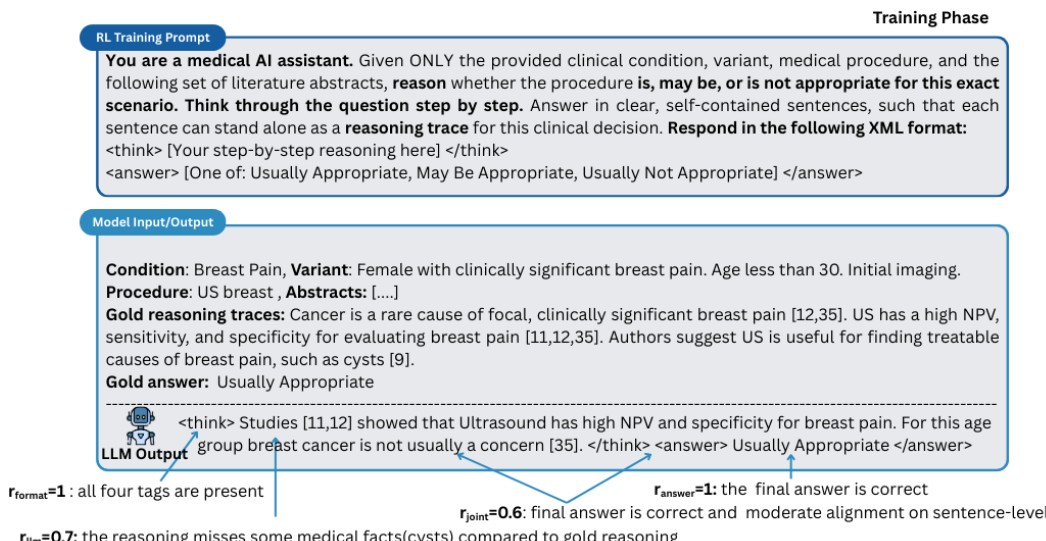

Figure 3: Example of training and respective rewards.

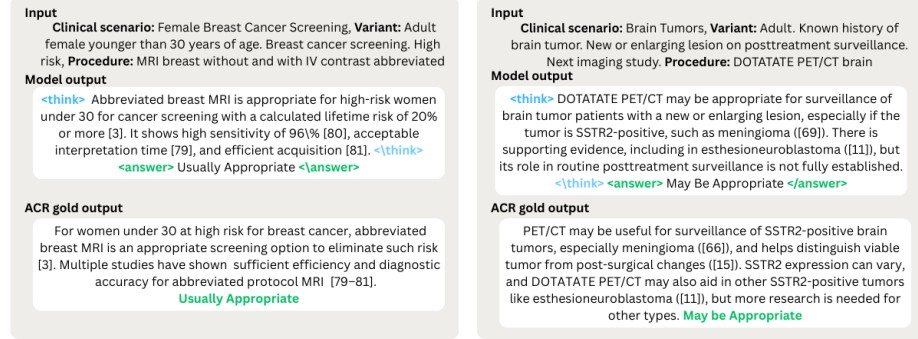

Figure 4: Example of model answers from MedReason-Embed Model

## A.4 REWARD TRAJECTORIES

As an example of training dynamics, we show the *MedReason-Embed* model that shows the healthiest curves, with a sharp reward jump after epoch 1 suggesting an "aha moment"; described by other studies as the point where the model "autonomously develops advanced problem-solving strategies, including reflection and self-correction" Guo et al. (2025). This is impressive given the limited training, as the model seems to move beyond simple pattern matching toward weighing clinical concepts like 'radiation risk' against 'diagnostic sensitivity,' much like an expert. It later plateaus around epoch 2 at about 1.4/2, where training was concluded.

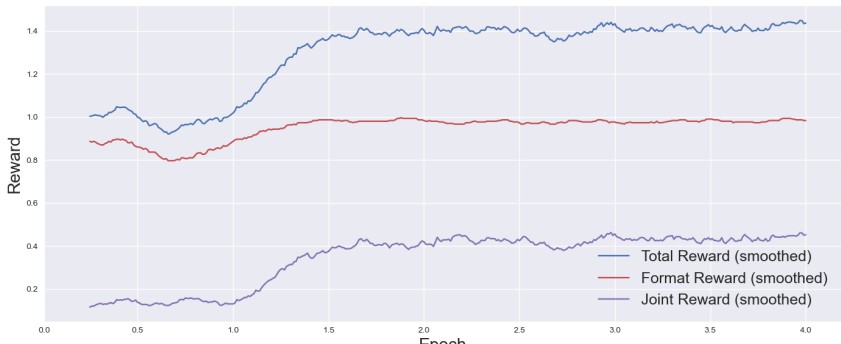

Figure 5: Smoothed training reward trajectories: reward components for each MedReason-Embed over 4 epochs, averaged with a window size of 25.

# B MEDICAL REVIEW AGENT WITH DEEPRETRIEVAL

Table 9: The suggested input queries to DeepRetrieval with an example for "Breast Pain" condition and Mammography procedure and a condensed sample of the rewritten queries from DeepRetrieval.

| Input Queries |
|---|
| [Breast Pain] |
| Diagnostic imaging for [Breast Pain] |
| Clinical evidence for the use of diagnostic imaging in the evaluation of [Breast Pain] |
| [Breast Pain] and related conditions |
| Affected conditions [Breast Pain] and synonyms |
| Other way to say [Breast Pain] |
| Clinical evidence for the use of [Mammography] in the evaluation of [Breast Pain] |
| P: [Breast Pain]; I: [Mammography]; C: Alternative imaging procedures; O: Diagnostic accuracy, risk and benefits |
| **Rewritten Queries** |
| ((Breast Pain OR Mastalgia) AND (Breast Cancer OR Breast Cancer Risk OR Cancer Risk) AND (Diagnostic Imaging)) |
| ((Breast Pain OR Mastalgia) AND (Imaging OR Mammography OR Magnetic Resonance Imaging OR Ultrasound OR Breast Imaging) AND (Diagnostic Imaging)) |
| ((Mammography OR mammography) AND (Breast Pain OR Mastalgia OR Mastodynia) AND (Diagnostic Imaging)) |

## C  POST-FILTERING AGENT

The Post-Filtering task is defined as predicting the *Strength of Evidence (SOE)* score for each retrieved study, using the ACR-AC 4-level scale (1 = low, 4 = high). The goal is to identify studies in the top SOE category, prioritizing those most relevant for clinical decision-making. This component was able to spot studies in the top of the 4 categories (high SOE) with 0.74 recall.

Table 10: Features for SOE Predictor

| 1. Publication Type | |
| --- | --- |
| Study Design | One-hot encoded (e.g., "Clinical Trial," "Review," etc.) |
| **2. GRADE Features (Extracted from Abstract)** | |
| Mentions patient outcomes | Binary; indicates whether the abstract refers to clinical or patient outcomes (e.g., mortality, morbidity). |
| Mentions accuracy metrics | Binary; indicates whether diagnostic accuracy metrics are reported (sensitivity, specificity, AUC). |
| Mentions comparator | Binary; indicates presence of a comparator, control group, or reference standard. |
| Mentions treatment or effect | Binary; indicates references to treatment, therapy, or impact on clinical management. |
| Mentions blinding | Binary; indicates whether blinding or masking was reported. |
| Mentions randomization | Binary; indicates whether the study employed randomization. |
| Sample size reported | Integer; the sample size if explicitly stated in the abstract (e.g., "n = ..."). |
| Mentions confidence interval | Binary; indicates whether confidence intervals are reported. |
| Mentions funding | Binary; indicates whether the abstract discloses funding sources, grants, or sponsorship. |
| **3. Journal and Year Features** | |
| SJR | Scientific Journal Rankings; metric of journal quality. |
| Year | Year of publication. |

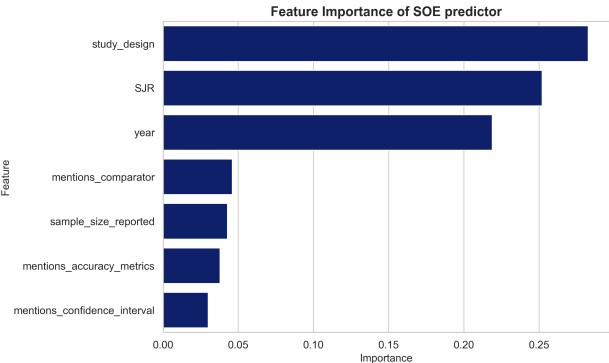

Figure 6: Feature importances observed using a Random Forest predictor

# D ICD CODING AGENT

Table 11: Examples of how different Italian clinical notes are mapped to ICD Code 185: Malignant neoplasm of prostate.

| ICD Code | Clinical note (Italian) | LLM-Standardized Clinical Note |
|---|---|---|
| 185 | tumori maligni della prostata | Malignant neoplasm of prostate |
| 185 | tumori maligni della prostata  integrazione richiesta specialistica | Malignant neoplasm of prostate |
| 185 | tumore maligno prostatico con sospetto metastasi | Malignant neoplasm of prostate with suspected metastasis |

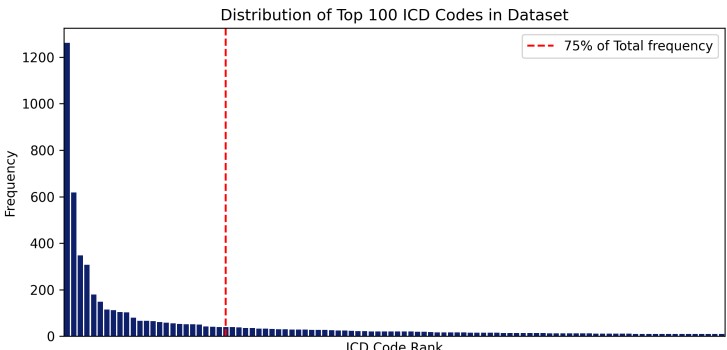

Figure 7: **Distribution of the top 100 ICD codes:** The red dashed line shows where cumulative frequency reaches 75% of occurrences (at code rank 25), highlighting the long-tail pattern seen in ICD distributions.

# E USE OF LARGE LANGUAGE MODELS (LLMS)

For this work, an LLM (GPT-5) was used solely for writing assistance, specifically to polish grammar, style, and clarity and to condense the main text. No LLMs were used for research ideation or methodological development. The authors take full responsibility for the content of this paper.

