# OpenReview forum: "Reinforcement Learning for Clinical Reasoning: Aligning LLMs with ACR Imaging Criteria"
_ICLR.cc/2026/Conference — Submitted to ICLR 2026_

### Official Review · Reviewer_wbkT · 2025-10-27

**Soundness:** 3
**Presentation:** 2
**Contribution:** 2
**Rating:** 4
**Confidence:** 5

**Summary:**

This paper introduces a modular, agent-based system for supporting clinical imaging recommendations by aligning large language models (LLMs) with American College of Radiology Appropriateness Criteria (ACR-AC). The approach centers on training a Reasoning Agent using Group Relative Policy Optimization (GRPO) and designing custom, reasoning-focused reward functions to encourage expert-like intermediate reasoning steps, rather than simply optimizing for correct final answers. The pipeline includes ICD-9 coding, multi-step retrieval and filtering of medical literature, and an end-to-end evaluation framework to test robustness, generalization, and scalability. Experiments demonstrate that the reasoning-augmented RL model (MedReason-Embed) delivers significant macro F1 improvements over standard supervised training and even outperforms much larger models, with added transparency and potential for generalization beyond static guideline scenarios.

**Strengths:**

1. Holistic Agentic Design: The modular, agent-oriented pipeline is thoughtfully engineered for practical and scalable deployment in real-world clinical imaging workflows. The inclusion of ICD coding and PubMed-based retrieval tightly integrates with established clinical data and processes, addressing both interoperability and evidence curation bottlenecks. Figure 2 is highly effective in illustrating system flow; it clarifies how outputs of each agent (e.g., ICD Coding Agent, Criteria Checker, Post-Filtering Agent) are chained, and what justifications pass through the system.
2. Originality in Reward Formulation: The exploration and systematic comparison of reasoning-focused reward functions, especially the “joint reasoning” cosine similarity-based MedReason-Embed reward (Eq. on Page 6 and Section 3.3), fills a prominent gap in medical AI, where prior work often settles for simple answer and format rewards. The linking of reasoning quality directly to outcomes represents a creative step toward more trustworthy AI in clinical domains.
3. Comprehensive Experimental Evaluation: The technical rigor of experimental protocol is commendable. The authors conduct extensive quantitative benchmarking (Tables 2–6), with well-chosen metrics (macro/weighted F1, LLM-alignscore, NER Embedding F1), fair baselines, ablations, and out-of-distribution tests on “generalization conditions." The side-by-side performance with SFT models and much larger models (LLaMA-3.1-405B) is transparently reported.
4. Methodological Rigor in Mathematical Details: The implementation of the GRPO objective is clearly stated, including the group relative advantage calculation and explicit KL penalty term in the loss (Equation in Section 3.3).
Impact and Reproducibility: Code availability and detailed hyperparameter settings (Appendix A), along with dataset statistics and pipeline explanations, support future reproducibility and foster open-source research.
5. Strong Use of Figures and Tables: Figures are well-integrated and informative. For instance, Figure 3 (Page 16) and Figure 4 (Page 16) provide clear, concrete examples of reasoning traces and how different model outputs are assessed and rewarded—this lends transparency to the reward formulation. Table 3 and Table 4 document the quantitative performance improvements across model variants and reasoning alignment scores, underpinning key claims.
6. Meaningful Generalization Experiments: The effort to test on conditions outside of the ACR-AC training set and compare pipeline-retrieved citations with gold-standard ACR references (Tables 5 and 6) is excellent and directly tied to the primary motivation for scalable, guideline-independent AI.
7. Interpretability and Clinical Trust: The stepwise, tag-structured outputs (<think>, <answer>), and ablation results detailing improvements in qualitative alignment, are likely to be of interest to clinicians seeking interpretable AI.

**Weaknesses:**

1. Several directly relevant recent works are not cited or compared—especially those advancing lightweight clinical reasoning LLMs, rule-based reward alignment, or multimodal decision support in radiology.
2. The core experimental dataset is limited to 1,800 triplets from just 30 ACR-AC conditions, with only 4 “unseen” generalization conditions for OOD evaluation. This is a very limited slice of the clinical imaging space (out of the 257+ topics available). As such, the demonstrated generalization is suggestive but not conclusively established. Broader or more diverse generalization studies (e.g., multiple medical centers, complex/ambiguous notes, higher ICD code diversity) are needed to make strong claims about real-world robustness.
3. Absence of In-Depth Theoretical Analysis of Reward Behaviors.
4. For evidence retrieval and filtering, the pipeline depends solely on PubMed abstracts and metadata, rather than utilizing full-text papers. This choice, though practical, risks missing nuanced evidence, study limitations, or trial context, which may undermine the strength of retrieved recommendations. While acknowledged under “Limitations,” no systematic impact analysis is performed—in particular, there are no ablation studies quantifying the performance drop or hallucination risk versus using full articles.
5. Although Macro F1 and Weighted F1 are appropriately reported in Tables 3–6, the manuscript does not present class confusion matrices or a deeper breakdown (e.g., per-condition or per-procedure breakdowns), which would help clarify error modes and class imbalance effects.
6. There is no user study or assessment of utility or trust by practicing clinicians, and therefore, while the approach is clinically motivated and outputs are interpretable, user acceptance is merely speculative at this point.

**Questions:**

1. How does the model handle ambiguous, multi-diagnosis clinical notes? Most of the ICD coding experiments are run on short, unambiguous cases (Table 2, Figure 7). How would the pipeline perform on real-world hospital notes with several possible ICD codes per case, and what adaptation strategies are envisioned for deployment?
2. What are the error modes of the MedReason-Embed reward? Can the authors provide failure examples where reasoning trace cosine similarity aligns with expert annotations but the underlying logic is flawed or leads to clinically invalid recommendations? Is there empirical evidence of the frequency and clinical significance of such cases?
3. Impact of Using Full-Text Evidence: Abstract-level reasoning may miss nuance and introduce bias. Have the authors tested retrieval or reasoning using full articles instead of only abstracts? If so, how do performance and hallucination risk change?
4. Generalization to Other Modalities or Guidelines: Since the approach is modular and claims extension potential, did the authors conduct any preliminary testing (even qualitative) on non-imaging guidelines, or is generalization purely hypothetical?
5. Training Stability and Efficiency: Table 4 and Figure 5 suggest differences in training efficiency (esp. LLM-Eval vs. MedReason-Embed). Can the authors share absolute wall times/hardware specs for main runs, and elaborate on bottlenecks or instability encountered, especially for larger batch/group sizes?

**Details Of Ethics Concerns:**

N/A.

---

> ### Author Response · Authors · 2025-11-27
> **Author response to reviewer wbkT.**
>
> We thank the reviewer for the detailed and constructive feedback. Below we clarify key decisions and planned extensions.
>
> Weaknesses
>
> 1. Related work
> We acknowledge the omission of several recent works on lightweight clinical LLMs and reward-shaping approaches. This was due to space limits and the fact that some papers appeared late during writing. We will expand the related-work section in the camera-ready to include RLHF-style medical alignment and compact clinical models. To our knowledge, none of these are trained explicitly on ACR Criteria, which remains a main novelty of our work.
>
> 2. Data generalization
> Training across more unseen ACR conditions was not feasible within our GPU budget at submission time. We agree this is an important extension and plan to expand the unseen-condition benchmark in the full study.
>
> 3. Reward behavior analysis
> We appreciate this suggestion. The core motivation was to test whether explicit “reasoning rewards” improve both justification quality and decision accuracy (see Sec. 2.2, lines 145–152). We also note in Limitations that reasoning rewards may miss certain clinical nuances. Now that we have empirical signals that reasoning helps, the next step is developing more clinically grounded rubrics, which we will include in a follow-up study.
>
> 4. Use of abstracts
> We used abstracts due to (1) context-length constraints and (2) paywalled full-text access. We agree that full-text retrieval could reduce hallucinations and improve nuance, and we plan ablations in a future version using larger-context models. Importantly, we observe substantial improvements when any evidence is provided vs. baseline, suggesting this direction is promising.
>
> 5. Lack of detailed breakdowns
> We agree this would clarify error modes. Due to page limits we omitted this analysis, but we will include per-condition/per-procedure F1 and confusion matrices in the camera-ready appendix.
>
> 6. No clinician study
> We agree that clinical validation is ultimately necessary. This work focuses on establishing a computational pipeline for reasoning-aligned RL. For the extended study, we plan blinded radiologist evaluation to assess reasoning-trace quality, reward calibration, and clinical acceptability, especially for out-of-guideline cases.
>
> Responses to Questions
>
> 1. Handling ambiguous clinical notes
> Our ICD experiments use short, single-diagnosis notes from real hospital data. ICD coding is known to be hospital-specific, and multi-diagnosis notes require different strategies. For such cases, we plan to use multi-label retrieval (top-k ICD codes with adaptive thresholding). This can be integrated easily in future work, but our current design reflects the dataset used.
>
> 2. Error modes of Reasoning-Embed
> Internal checks show the case where correct reasoning leads to an incorrect label occurs mainly early in training, before the model learns to integrate risk-benefit comparisons. These instances were rare and decreased over time. This is one motivation for multiplying reasoning reward by final-answer correctness. We will include a short failure-case analysis in the appendix.
>
> 3. Abstract-only context
> As noted above, abstracts were used for compute and access reasons. We acknowledge missing nuances may limit performance. Future work includes building a parsed full-text database or using an LLM summarizer to extract key findings. Despite this limitation, adding abstract-level evidence already yields large gains compared to the no-context baseline, which is a significant insight
>
> 4. Generalization to other guidelines/modalities
> We agree this is a valuable generalization benchmark. Performing transfer evaluations on other topics is planned for the extended version, particularly since guideline-invariant reasoning is one of the intended benefits of our process-supervised approach. Something we will test once we train with all the conditions in the ACR. We will not say that “generalization is purely hypothetical”.
>
> 5. Training stability and efficiency
> All experiments ran on a single A100 80GB. End-to-end wall-clock time for the baseline GRPO model was ~2 hours (details in Appendix A1 and Table 4). The main bottleneck was reward computation, not GRPO optimization.
> – LLM-Eval incurred high latency due to evaluator calls per generation.
> – Embedding rewards were more efficient and stable but still required embedding each generated sentence.
> Training was stable across all runs with LoRA rank 32, paged_adamw_8bit, batch size 6 (grad accumulation 2), and smooth loss curves.

---

### Official Review · Reviewer_hJMK · 2025-10-29

**Soundness:** 2
**Presentation:** 2
**Contribution:** 3
**Rating:** 4
**Confidence:** 3

**Summary:**

This paper offers a multi-agent system designed to help LLMs adhere to the American College of Radiology's imaging guidelines. The authors are tackling a significant real-world problem. these evidence-based guidelines are widely underutilized by clinicians. The proposed system automates the entire imaging referral workflow, starting from interpreting a clinical note and assigning an ICD code, then retrieving relevant evidence, and finally producing a reasoned recommendation. The core of this work is the Reasoning Agent, a lightweight 8B model trained using GRPO. The key innovation is its process-oriented reward function, which simultaneously optimizes for a correct final answer and for semantic alignment with expert-derived reasoning steps. The results show that MedReason-Embed model can outperform both standard supervised fine-tuning and even a much larger 405B parameter model.

**Strengths:**

- This work addresses a highly significant and practical clinical problem. Reducing the overuse of low-value imaging has direct, positive implications for both patient safety (e.g., lower radiation exposure) and healthcare costs.

- The modular, multi-agent architecture is a very sound engineering choice. It intelligently decomposes an extremely complex clinical workflow into distinct, inspectable components (coding, filtering, reasoning), which is a practical approach for future development and real-world integration.

- The paper’s focus on process-supervised reinforcement learning is a valuable methodological contribution. Moving beyond simply rewarding the final correct answer to instead scrutinize the intermediate reasoning steps is a crucial direction for building more transparent and trustworthy medical AI.

- Showing that a carefully tuned 8B model can achieve stronger results than a massive 405B model is a powerful finding. It clearly demonstrates the high value of sophisticated, domain-specific reward engineering over a brute-force approach of simply scaling up model size.

**Weaknesses:**

- My primary concern is the claim of generalization, which I find to be critically underdeveloped. The entire claim of robustness to out-of-distribution cases rests on a test set of only four unseen conditions. This is simply not a large enough sample to draw any meaningful conclusions or to support such a broad claim.

- The paper is missing the single most important validation step, an evaluation by human clinical experts. This is a system built for clinical decision support, yet no radiologists or physicians were brought in to review the final recommendations. As a result, we have no data on the clinical validity, safety, or real-world utility of its outputs.

- The reasoning-focused reward feels superficial. It relies on semantic similarity (cosine similarity) to a gold trace, which is not a reliable proxy for logical or clinical correctness. It seems entirely possible for this metric to reward a model for generating text that sounds correct but is clinically flawed or even dangerous.

- The evidence-gathering pipeline relies exclusively on abstracts, not full-text articles. This is a significant flaw, as abstracts are summaries and frequently omit the critical methodological limitations, data nuances, or contraindications that are essential for making a sound medical judgment.

- The presentation quality of the figures and tables also detracts from the paper. Several key visuals intended to clarify the methodology, such as the system architecture diagram, are instead vague and fail to clearly illustrate the complex data flow between agents. Furthermore, some crucial tables reporting main results and features suffer from poor formatting, making them difficult to read and forcing the reviewer to hunt for the information.

**Questions:**

- Could you provide a much stronger justification for your generalization claims? For instance, what was the clinical heterogeneity of those four test conditions, and how do they represent a wider class of problems?

- Were any human clinicians or radiologists involved at any stage, either in validating the "gold" reasoning traces you extracted or, more importantly, in reviewing the final recommendations from your MedReason-Embed agent?

- How does your embedding-based reward function penalize reasoning that is semantically similar to the gold trace but contains a critical logical flaw or clinical error?

- Given the system's reliance on abstracts, how do you propose to handle the inevitable cases where the critical piece of information (e.g., a key limitation, a specific patient subgroup) is only mentioned in the full text of a study?

---

> ### Author Response · Authors · 2025-11-27
> **Author Response to Reviewer hJMK**
>
> We thank the reviewer for the thoughtful and detailed feedback. Below we address each question and outline improvements planned for the camera-ready version.
>
> 1. Generalization justification
> We agree this needed clearer explanation. The four held-out conditions were intentionally chosen to be clinically heterogeneous relative to the 30 training conditions, varying along:
> – organ system (e.g., ovarian cancer screening vs. breast pain),
> – presentation type (acute vs. chronic), and
> – reasoning style and evidence patterns (empirically confirmed via inspection of ACR justifications).
>
> This diversity was chosen explicitly to test whether the model can handle shifts in clinical context and reasoning style; even with a small number of held-out tasks. We fully acknowledge that evaluating only four generalization conditions is a limitation, and expanding beyond these four was not possible at this stage of the project, but we believe to have noted this in our Limitations. In our subsequent paper (already in the works) we aim to include more conditions.
>
>
> 2. Clinician involvement
> No practising clinicians or radiologists were prospectively recruited in this study. This is a limitation of the current study, which we will state explicitly.
> However, we believe that clinical expertise is embedded in:
> - Gold reasoning comes from expert ACR-AC documents, not from our models. These are written by multidisciplinary expert panels following formal evidence-review and GRADE methodology. Our extraction step produces shorter, atomic claims but preserves their expert clinical content.
>
> Additionally the scope of this work was to demonstrate a computational reasoning-alignment framework and show feasibility (improved accuracy and reasoning alignment, and generalization to unseen conditions). All evaluations are retrospective and guideline-based; we do not claim clinical readiness.
>
> In the camera-ready, we clarify that (i) ACR documents serve as expert supervision; (ii) a blinded clinician study is planned for future work.
>
> 3. Penalty for flawed but semantically similar reasoning
> That is a very good point and we agree with the reviewer: MedReason-Embed does not detect all logically or clinically flawed reasoning that remains embedding-similar to expert traces. This is a known limitation of embedding-based alignment and is already noted in our Limitations.
>
> The reward behaves as follows:
> – If the final label is wrong, reward = 0 (so flawed reasoning that leads to an incorrect decision is never reinforced).
> – If the label is correct, reward encourages semantic similarity to expert traces, improving concept overlap (NER-F1) and LLM-judged reasoning (Table 4).
>
> Residual failures remain e.g., answer-correct but subtly flawed reasoning that stays close in embedding space. We view MedReason-Embed as a computationally lightweight first step; richer clinical reward rubrics (explicit logic checks, radiologist-defined key criteria, or an LLM critic trained to flag unsafe reasoning) are natural next extensions. We will clarify this explicitly.
>
> 4. Handling information only present in full-text articles
> This is also a very good point and we fully agree that abstract-only retrieval can miss clinically important details. We used abstracts due to (i) context-length limits and (ii) full-text licensing constraints. This is a deliberate, conservative design choice in this first study.
>
> As a consequence:
> – The model may miss recommendations dependent on fine-grained, full-text-only information.
> – Therefore we believe that given sufficient computational resources and LLMs with bigger context sizes, our results can be seen as a conservative lower bound on performance achievable with richer evidence, assuming that the LLM can correctly identify them.
> Despite this, adding abstract evidence already yields substantial gains over no-context baselines; our main empirical insight.
>
> Future directions can also include hierarchical retrieval: use abstracts for initial retrieval, then fetch full text for top-k studies and supply section-aware summaries to the model or even uncertainty flagging: for conditions where abstract evidence is insufficient, explicitly indicate the need for full-text review rather than hallucinating confidence.
>
> In the camera-ready version, we will clarify these points in the Limitations and Future Work sections, emphasizing that:
>  (a) abstract-only retrieval is a deliberate, conservative design choice for this first study;
>  (b) it likely underestimates the achievable performance; and
>  (c) the methods we propose (hierarchical retrieval and explicit escalation when evidence is incomplete) are natural next steps for incorporating full-text information.

---

> > ### Comment · Reviewer_hJMK · 2025-11-27
> >
> > Thank you to the authors for the detailed comment. I will retain my score.

---

### Official Review · Reviewer_kSxk · 2025-11-01

**Soundness:** 3
**Presentation:** 3
**Contribution:** 3
**Rating:** 6
**Confidence:** 3

**Summary:**

The paper proposes an **agentic clinical decision-support pipeline** that aligns LLM reasoning with the **ACR Appropriateness Criteria (ACR-AC)** via **reinforcement learning**. The system comprises: (i) an **ICD coding agent** that maps clinical notes to ICD-9-CM codes; (ii) a **Reasoning Agent** trained with **Group Relative Policy Optimization (GRPO)**; (iii) a **medical evidence retriever** (DeepRetrieval-PubMed-3B) with a **post-filtering module** approximating GRADE-style strength-of-evidence scoring; and (iv) an end-to-end controller that mirrors ACR guideline workflow.

Key technical ideas:
- **Reasoning-aligned RL**: multiple reward designs compare generated “reasoning traces” against expert rationales distilled from ACR-AC (“Baseline”, “Citations”, “LLM Eval”, and **MedReason-Embed** joint reward). (*Section 3.3; Table 1 on p.6*).
- **Modular agent architecture** grounded in guideline workflow. (*Figure 2 on p.4*).
- **Evaluation** on ~1,800 triplets from 30 ACR conditions (70/30 split), plus **four unseen conditions** for generalization; metrics include **Macro/Weighted F1**, an **LLM-based alignment score**, and **NER-based F1** for reasoning overlap. (*Section 4; Tables 3–6 on pp.7–8*).

Main results:
- **MedReason-Embed** and **LLM Eval** outperform SFT and a larger unfine-tuned LLaMA-3.1-405B on Macro/Weighted F1 and reasoning alignment, at lower or comparable compute than LLM-as-judge rewards. (*Tables 3–4*).
- The ICD agent achieves **Top-1 80.45%** and **Hierarchical Top-1 91.47%** accuracy on a synthetic/curated dataset. (*Table 2 on p.7*).
- Under distribution shift (unseen conditions) and with non-ACR citations, performance degrades modestly, preserving model ranking. (*Tables 5–6 on p.8*).

Overall, the paper argues that **process-supervised RL** (reasoning rewards) plus **retrieval grounding** yields **more trustworthy** and **generalizable** clinical imaging recommendations than SFT alone. :contentReference[oaicite:0]{index=0}

**Strengths:**

1. **Well-aligned system design** that mirrors real guideline workflows; compelling **architecture** with clean module boundaries.
2. **Reasoning-first RL**: clear comparison of **answer-only vs. reasoning-linked** rewards; the **MedReason-Embed** joint reward is simple and effective.
3. **Grounded retrieval and filtering**: DeepRetrieval query rewrites and GRADE-inspired post-filtering with interpretable features.
4. **Transparent reporting** of training config, reward examples, and trajectories (suggestive “aha” phase).
5. **Practical ICD mapping pipeline** with decent accuracy and a sensible metric suite (Top-k, hierarchical, MRR).

**Weaknesses:**

1. **Limited scale & diversity**: 30 training conditions and 4 unseen conditions are **too narrow** to support strong generalization claims across ACR’s 257 topics; add ≥50–80 unseen scenarios across multiple body systems.
2. **Reasoning metric fidelity**: LLM-as-judge and NER-overlap can reward fluent but shallow chains or penalize correct-but-paraphrased reasoning. Consider **expert rubric scoring** and **counterfactual tests** (e.g., sensitivity to omitted risk factors).
3. **Potential leakage**: since rewards and test rationales both derive from ACR artifacts, the setup may favor **template-matching**; adopt **held-out authoring** or **cross-guideline** transfer (e.g., NICE/ESR) to evidence real reasoning transfer.
4. **Safety & governance**: no human evaluation by radiologists here; prospective trials or **reader studies** are needed before clinical claims.
5. **ICD-9 reliance**: demonstrate portability to **ICD-10/ICD-11** and non-English clinical notes beyond the curated Italian dataset; quantify effects of mapping errors on downstream decisions.
6. **Statistical reporting**: while McNemar’s test is noted, the paper should provide **per-class F1**, **confidence intervals**, and **multiple-run variance** (seeds) for RL models.

**Questions:**

1. **Generalization breadth**: Can you expand the unseen-condition set to ≥50 topics spanning trauma, oncology, pediatrics, and cardiovascular domains? What is the expected performance drop vs. in-domain?
2. **Robustness tests**: How do models behave under **evidence ablation** (remove key trials) or **contradictory abstracts**? Does MedReason-Embed still choose safe imaging?
3. **Reward analysis**: For MedReason-Embed, how sensitive are results to sentence segmentation and embedding choices? Any **failure cases** where correct answers get down-weighted for paraphrases?
4. **Human evaluation**: Any preliminary **radiologist blinded review** (appropriateness + justification) on a random stratified sample?
5. **Cross-guideline transfer**: If trained on ACR, how well does the system perform on **NICE/ESR** topics without retraining?
6. **Coding standards**: What is required to migrate to ICD-10/ICD-11, and how does coding accuracy affect end recommendation error rates?
7. **Safety guardrails**: Are there hard constraints (e.g., always down-weight CT for pregnancy unless specific red flags) to prevent reward gaming or hallucinated justifications?

---

> ### Author Response · Authors · 2025-11-27
> **Author response to reviewer kSxk**
>
> We thank the reviewer for the detailed and constructive feedback. Below we address each point and outline the concrete changes for the camera-ready version.
>
> 1. Generalization breadth
> At the time of writing, expanding to ≥50–80 unseen conditions across multiple ACR body systems was not feasible due to computational constraints. However, we fully agree this is an important extension, and expanding the unseen-condition (generalization set) benchmark is a priority for the full study version of this work.
>
> 2. Robustness tests
> We would argue that the “Baseline” model is the “no-evidence model” which implicitly captures the “key trials removed” scenario, and as it was seen this model did not perform as well compared to the contextualized models.
> Testing performance under contradictory abstracts is indeed a valuable robustness check, especially for safety, and we can incorporate this testing. So far though the only safety check is for “high quality” abstracts which aims to make sure that the paper is of high quality, assuming that its conclusion are the “correct ones and clinically safe”.
>
> 3. Reward analysis for MedReason-Embed
> We tested three different embedding models from HuggingFace; results were stable across choices.
> We agree embeddings are not ideal for reasoning fidelity, but they provide a close-enough signal; we argue that based on our initial checks there were no failure cases from paraphrasing, on the contrary the multiplication (embedding x correctness reward) was highly innovative and actually prevented rewarding semantically similar reasoning that leads to an incorrect conclusion.
> Fine-tuning domain-specific embeddings is also a promising next step, or finding other ways and frameworks to more accurately evaluate the reasoning.
>
> 4. Human evaluation
> We have not yet conducted a blinded radiologist review. This is something we wish to include in the larger study, as clinical validation is essential before deployment. This can be both to train a reward model for clinical reasoning but also for the final evaluation metrics (developing rubrics etc).
>
> 5. Cross-guideline transfer
>
> We agree this is a valuable generalization benchmark. Performing transfer evaluations on other topics is planned for the extended version, particularly since guideline-invariant reasoning is one of the intended benefits of our process-supervised approach. Something we will test once we train with all the conditions in the ACR.
>
> 6. ICD-10/ICD-11 migration
> We would anticipate that migration requires minimal changes. The choice of ICD-09 was mainly because of our dataset. Because the coding agent is retrieval-based and modular, replacing ICD-9 with ICD-10/ICD-11 involves swapping the description corpus, without altering downstream components. It is worth noting that ICD-9 is one of the most used ones globally, further motivating our choice.
>
> 7. Safety guardrails
> Explicit rule-based guardrails (e.g., pregnancy→avoid CT unless red flags) are not yet implemented. However, we argue the reliance on high-quality abstracts (post-filtering agent function) already filtered by study design and strength; implicitly enforces some degree of safety.
> Adding explicit constraints can be part of our planned safety work and they need to be done with medical experts.
>
> 8. Statistical reporting
> We appreciate the suggestion. In the camera-ready, we will include per-class F1 scores and additional statistics in the Appendix.

---

### Official Review · Reviewer_3vg1 · 2025-11-01

**Soundness:** 2
**Presentation:** 2
**Contribution:** 2
**Rating:** 2
**Confidence:** 3

**Summary:**

The manuscript presents a reinforcement learning (RL) approach for clinical reasoning, aiming to align large language models (LLMs) with the ACR Imaging Appropriateness Criteria. The authors propose an end-to-end agentic architecture that integrates ICD coding, evidence retrieval, and a reasoning agent trained with Group Relative Policy Optimization (GRPO). While the topic is relevant and the proposed method has potential, the manuscript suffers from significant issues in presentation and coherence that hinder its readability and academic rigor.

**Strengths:**

The work aims to address a significant clinical problem—the reduction of unnecessary medical imaging—by leveraging AI to implement evidence-based guidelines. This alignment with a real-world healthcare challenge is a clear strength.

**Weaknesses:**

1. The manuscript reads as a collection of loosely connected sections. Key components—such as the ICD Coding Agent, Reasoning Agent, and evidence retrieval—are described in isolation, with insufficient transitions to tie them together.
2. The methodology section lacks a clear motivation. For instance, the choice of GRPO over other RL methods is not adequately justified, and the relationship between different reward functions and their impact on clinical reasoning remains unclear.
3. The results are presented in a fragmented manner, making it difficult to draw meaningful conclusions about the overall effectiveness of the proposed system.
4. The description of the reasoning trace extraction process is vague. The use of `LLaMA-4-Scout-17B-16E-Instruct` is mentioned, but the rationale for selecting this model and the steps taken to prevent hallucinations are not sufficiently detailed.
5. The reward functions (e.g., `MedReason-Embed`) are introduced without a strong theoretical or empirical justification. The link between reasoning quality and clinical validity is assumed but not thoroughly argued.

Minor issues:

1. Figures such as Fig. 1 and Fig. 2 are pixelated and blurry when enlarged, making it difficult to interpret key components of the proposed system architecture and guideline development process. High-resolution, vector-based figures are essential for clarity and reproducibility.

2. The writing contains informal phrases and ambiguous terminology (e.g., “reasoning-focused rewards,” “lightweight model”), which should be replaced with more precise language.

**Questions:**

see above.

---

> ### Author Response · Authors · 2025-11-27
> **Author response to Reviewer 3vg1**
>
> We thank the reviewer for the thoughtful and detailed feedback. Below we address each point and outline improvements planned for the camera-ready version.
>
> 1. “The manuscript reads as a collection of loosely connected sections.”
>
> The modular structure reflects the modular agent-based architecture: Section 3.1 gives the full pipeline overview, while Sections 3.2–3.4 isolate each agent for clarity and reproducibility. This was a design choice to ease the reader but we agree transitions can be strengthened. In the camera-ready, we will add short bridging statements at each subsection start to make upstream–downstream dependencies clearer while keeping the modular layout.
>
> 2. Motivation for GRPO and reward choices
>
> We agree that we can make the motivation more explicit. To clarify: GRPO was chosen because (i) it is the standard in recent medical RL works (MedR1, MedVLM-R1), enabling direct comparability; (ii) it removes the critic network, reducing memory footprint( important in clinical settings); (iii) it is well suited for sparse, sequence-level rewards, unlike PPO’s token-level value estimates; and (iv) its relative-advantage formulation fits our goal of aligning reasoning quality.
>
> Regarding the reward choices:
> We believe the relationship is described, though we agree it can be made more prominent. Table 1 shows how each model incrementally adds (i) evidence context, (ii) LLM-based reasoning evaluation, and (iii) our joint embedding reward. Section 3.3 (lines 267–288) describes each reward's clinical rationale: the answer reward ensures correct recommendations; the format reward enforces structured output for interpretability; the LLM Eval reward teaches what constitutes clinically sound reasoning by comparing to expert traces; and our MedReason-Embed reward (Equation in line 288) explicitly couples reasoning quality to outcome correctness; addressing the known failure mode where models produce correct answers via flawed reasoning (Chen et al., 2025). This can also be described more in theoretical depth in the camera-ready.
>
> 3. Fragmented results
>
> The results are structured per agent because each module has distinct evaluation criteria, but we agree connections between sections can be clearer. We will add explicit linking sentences summarizing how each module’s performance contributes to overall system effectiveness.
>
> 4. Reasoning-trace extraction and hallucination control
>
> We thank you for your comment regarding the choice of LLM for reasoning extraction. LLaMA-4-Scout was selected due to strong structured-summarization performance, empirical tests showing fewer omissions/hallucinations vs. alternatives, and full reproducibility. We will add these details and an Appendix figure illustrating the extraction pipeline with examples.
>
> For the extraction process: we agree this should be clearer. Extraction was constrained summarization from ACR expert documents, not generative reasoning.  This is fundamentally different from (and far less hallucination-prone than) open-ended generation. The limitation of hallucinations was done 1) empirical testing of the backbone LLM (see above) 2) human oversight to a number of claims. In the future this validation process can also be helped with expert radiologists.
>
>
>
> 5. Theoretical/empirical justification of MedReason-Embed
>
> Our goal is to test whether reasoning-aligned rewards improve both reasoning fidelity and clinical decision accuracy (Sec. 2.2). MedReason-Embed follows standard PRM assumptions: correct clinical reasoning should co-occur with correct decisions. The multiplicative design (correctness × reasoning-similarity) avoids rewarding superficially good but clinically wrong reasoning. We also explicitly acknowledge in the Limitations section that reasoning-based rewards may miss certain logical or clinical nuances. Now that we have empirical evidence reasoning rewards can help, a natural next step is designing more clinically grounded criteria and rubrics, which we aim to do in a following study.
>
> 6. Minor issues
>
> – Pixelated figures: all graphics will be replaced with high-resolution vector images.
> – Informal phrasing: we will revise ambiguous expressions (“reasoning-focused rewards,” “lightweight model”) to more precise terminology.

---

### Meta-Review · Area_Chair_7nMn · 2026-01-07

**Summary:**

The reviewers were unconvinced on the positive side, with one reject, two marginally below the acceptance threshold and one marginally above the acceptance threshold. They agreed that this work requires additional effort to meet the acceptance bar of ICLR. Thus, I am inclined not to accept this draft at this stage. Thank you for your effort. It is an interesting work. I hope the input from the reviewers will help you further improve this work.

**Reviewer Concerns:**

This work has limited novelty and experimental results.

**Reviewer Scores:**

The reviewers' scores reflect the limitations of this work.

---

### Decision · Program_Chairs · 2026-01-26

Reject